# Auto-Segmentation and Quantification of Non-Cavitated Enamel Caries Imaged with Swept-Source Optical Coherence Tomography

**DOI:** 10.3390/diagnostics13233586

**Published:** 2023-12-03

**Authors:** Tamer Abdelrehim, Maha Salah, Heather J. Conrad, Hooi Pin Chew

**Affiliations:** 1Department of Restorative Sciences, University of Minnesota, 8-440 Moos Tower, 515 Delaware St. SE, Minneapolis, MN 55455, USAconr0094@umn.edu (H.J.C.); chew0014@umn.edu (H.P.C.); 2Department of Restorative Dentistry, Faculty of Dentistry, Ain Shams University, Cairo 11566, Egypt

**Keywords:** OCT, SEM, image segmentation, carious, batch processing, simulation, dental imaging, demineralization, quantification

## Abstract

(1) Background: OCT imaging has been used to assess enamel demineralization in dental research, but it is not yet developed enough to qualify as a diagnostic technique in clinics. The current capabilities of most commercial acquisition software allow for visual and qualitative assessments. There is a need for a fast and verified batch-processing algorithm to segment and analyze demineralized enamel. This study suggests a GUI MATLAB algorithm for the processing and quantitative analysis of demineralized enamel. (2) Methods: A group of artificially demineralized human enamels was in vitro scanned under the OCT, and ROI frames were extracted. By using a selected intensity threshold colormap, *Inter*- (*Ie*) and *Intra*- (*Ia*) prismatic demineralization can be segmented. A set of quantitative measurements for the average demineralized depth, average line profile, and integrated reflectivity can be obtained for an accurate assessment. Real and simulated OCT frames were used for algorithm verification. (3) Results: A strong correlation between the automated and known Excel measurements for the average demineralization depth was found (R^2^ > 0.97). (4) Conclusions: OCT image segmentation and quantification of the enamel demineralization zones are possible. The algorithm can assess the future development of a real-time assessment of dental diagnostics using an oral probe OCT.

## 1. Introduction

In dental caries research, there is a crucial need to be able to study demineralization with high-resolution tomographic imaging techniques that can be used both in vitro and in vivo, especially intraorally. One of the important aspects of enamel demineralization studies is the design of preventive and remineralization materials, which requires an understanding of the temporal and spatial development of distinct ultrastructural early demineralized enamel zones as well as the diffusion–reaction mechanisms of the de- and remineralization processes [1,2,3,4,5,6]. Early demineralized enamel has two layers, according to previous studies: *Intra*- and *Inter*-prismatic demineralization. *Intra*-prismatic demineralization is the first layer to be observed after the enamel surface; *Inter*-prismatic demineralization is the second layer, and healthy or sound enamel (*sound*) sits beyond this demineralization progression [7,8,9,10]. Transmission electron microscopy (TEM) and scanning electron microscopy (SEM) are the key imaging tools used to visualize and study these demineralization zones. However, the challenging aspects of using these microscopy techniques, such as their destructive natures and the intricate sample preparation processes, limit their uses to a small sample size and make monitoring temporal changes of the demineralization zones in vitro and in vivo impossible.

It has been noted that demineralized enamel or dentin possess altered optical characteristics compared to their *sound* statuses [11,12]. It is also noteworthy that at the beginning of the enamel caries process, the scattering coefficient of dental enamel increases considerably with increasing mineral loss. Substantial associations between mineral loss and near-infrared (NIR) scattering have been established [12,13], and optical coherence tomography (OCT) with a central wavelength of 1300 nm has been found to be particularly useful in assessing non-cavitated enamel demineralization because of its ability to provide non-invasive, depth-resolved, high-resolution near-infrared backscatter characteristics of demineralization. Moreover, the oral handheld OCT is considered to be extremely safe clinically compared to radiographs and gives more information in just one 3D scan. As a result, one area of OCT usage is the early diagnosis of enamel defects using the scattering coefficient. Research on the evaluation of the demineralization of dental hard tissue using OCT started more than 20 years ago [12,13,14,15,16,17,18,19,20,21,22]. OCT’s ability to detect minute cross-sectional optical and morphological changes in dental hard tissues without using ionizing radiation makes it advantageous, especially clinically.

Researchers have validated the qualitative distribution and pattern of OCT reflectance for fissure and smooth surface enamel caries against histological techniques such as polarized microscopy and confocal laser scanning microscopy and found good correlations [20,23,24] between them. Recently, a few studies have even reported using deep learning to classify these OCT reflectance distributions and patterns to detect caries [25,26]. In addition to utilizing the OCT reflectance pattern of enamel demineralization qualitatively, many quantitative outcome measures have been derived from the OCT depth-resolved reflectance profile in an attempt to quantify the severity of demineralization in terms of both the degree and depth of demineralization. The lesion depth, integrated reflectivity, and attenuation coefficients are among the quantitative outcome measures that are often utilized [18,19,20,21,27,28]. The relative changes of these outcome measures for enamel and dentin before and after an intervention, be it demineralization or remineralization, can be used to assess the effect of that intervention [29,30]. However, variations in these quantitative results between different studies have been found, and this has been attributed to the different OCT systems used, sample preparation methods, and the arbitrary nature of some of the outcome measures [31]. Demineralization, or lesion depth, is a much-favored outcome measure in most tomographic methods, such as microCT and confocal microscopy, as it impacts clinical treatment decision making. Consequentially, the same outcome measure has been explored for OCT reflectance. However, as the intensity range of OCT images is relatively wide, choosing an intensity threshold for the end point of a lesion proves to be a challenge. Le et al. [32] measured the lesion depth using edge-finding algorithms, while Can et al. [33] chose the intensity threshold for the lesion depth as 1/e^2^ times that of the peak intensity. Other studies [34,35,36] determined the lesion severity by integrating the reflectance of the OCT line profiles from the tooth surface to a selected depth. The integrated reflectance, however, can be affected by surface specular reflectivity and, in some cases, the speckle noise combined with OCT scans. Therefore, it has been suggested that there is a need for more standardized and robust OCT methodologies to diagnose and quantify early enamel caries lesions [31], which include improvements to the OCT scanning hardware and the optimization of the data extraction and image processing processes.

Different image processing methods have been explored for dental-related OCT reflectance, including an averaged intensity difference detection algorithm for gingival sulcus [37], a depth intensity profile analysis for the detection of microdamage after implant insertion [38], and an intensity-based layer segmentation algorithm for the detection of enamel abrasion and wear [34], to name a few. These algorithms could potentially be adapted for the analysis of demineralized enamel and dentine. It has also been suggested that deep learning algorithms may more precisely segment demineralized areas [39]. However, the algorithms do not yet distinguish between *Inter*- (*Ie*) and *Intra*- (*Ia*)prismatic demineralization and have the limitation of manual verification even when using microCT as a reference.

For longitudinal de- and remineralization studies and studies with large sample sizes, robust and targeted image processing methods such as the segmentation of demineralization zones are needed to minimize manual and fatigue errors and subjectivity during data extraction. This entails the need for a validated and streamlined data processing algorithm that is specific to a demineralized tooth structure. In order to validate data processing algorithms, the modulation and simulation of OCT signals have previously been carried out using Lab VIEW and MATLAB scripts [40].

This work reports on a verified automated algorithm to segment and measure the depth of *Intra*- and *Inter*-prismatic demineralization zones in early demineralized enamel scanned using swept-source optical coherence tomography (SS-OCT) images. This segmentation algorithm is based on intensity thresholding that was previously identified using SEM [41].

In this study, no speckle noise removal was used before the segmentation process of the demineralized zones. Speckles are caused by the interference of coherent light waves that are scattered by the tissue microstructure. Those scattered light waves can contain information about the tissue properties, such as the size, shape, and distribution of the scatterers. This type of noise affects the quality and contrast of optical coherence tomography (OCT) images [42]. Speckle reduction is a technique that aims to improve the image quality and visibility of the tissue features by suppressing the speckle noise. However, speckle reduction can affect the demineralization intensity in OCT images in different ways depending on the type and level of speckle reduction applied [43,44]. It can also alter the optical properties of the tissue and introduce artifacts or distortions in the OCT images. The speckle reduction can also affect the demineralization intensity in OCT images. Moreover, it can reduce the contrast and dynamic range of the OCT images, which can make it harder to distinguish between different levels of demineralization. Speckle reduction can smooth the edges and boundaries of the tissue structures, which can affect the accuracy and precision of the demineralization measurements. Speckle reduction can introduce bias or errors in the estimation of the optical properties of the tissue, such as the refractive index, scattering coefficient, and attenuation coefficient. These errors in the tissue’s optical properties can affect the demineralization intensity calculation. Therefore, speckle reduction is a trade-off between improving the image quality and preserving the tissue information in OCT images [45].

## 2. Materials and Methods

### 2.1. Sample Preparation and Optical Coherence Tomography Scanning Protocol (OCT)

Eight human extracted upper premolars with visually caries-free buccal surfaces were collected and cleaned. The crown of each tooth was then painted with an acid-resistant varnish except for approximately a 3 × 3 mm^2^ window of the buccal surface. Each specimen was then subjected to a different number of daily cycles of de-and remineralization, ranging from 14 to 21 days (*n* = 1).The modified Featherstone pH cycling model was used [46], whereby the daily pH cycling routine consisted of demineralization for 6 h in 40 mL of acid buffer (Millipore Sigma, Germany) (37 °C) containing 2.0 mmol/L Ca, 2.0 mmol/L PO_4_, and 0.075 mol/L acetate at pH 4.3, and then rinsed with deionized water and remineralized for 18 h (37 °C) in 20 mL solution containing 1.5 mmol/L Ca, 0.9 mmol/L PO_4_, 0.15 mol/L KCl, and 20 mmol/L cacodylate buffer at pH 7.0.

The swept-source optical coherence tomography, SANTEC IVS-2000 SS-OCT, with a central wavelength of 1300 nm, and the SANTEC Inner Vision IVS-2000 software were used to acquire the 3D OCT scans in this study. The OCT scanning protocol is described as follows: the 3 × 3 mm^2^ demineralized region of interest was centralized in a scan area of 5 × 5 mm^2^. The vertical distance of the specimens from the imaging probe was standardized with the highest point of the surface curvature positioned approximately at the 0.5 mm level of the SANTEC Inner Vision IVS-2000 imaging window. The scanning beam was oriented perpendicular to the buccal surface and configured to scan an area of 5 mm × 5 mm in the x-y direction and a depth of 1.85 mm in the enamel (refractive index = 1.63). The x-y-z pixel count was 256 × 256 × 1000, resulting in lateral and axial resolutions of 19.5 and 4.4 μm, respectively. In OCT scans, the swept source laser beam incident and scans on the test sample and the scattering beams are reflected to the OCT and then interfere with the beams reflected by its reference mirror, where different intensities for an object under the OCT scan indicate different scattering properties for each type of tissue. A scanning electron microscope (SEM) (JEOL 6500 Field Emission Gun SEM operating at an accelerating voltage of 5 kV) is used for some selected sections on the same tested demineralized enamel samples after the normal sample preparation for SEM scans to build up a correlation between the OCT and SEM frames that will be used to determine our specific color map for OCT frame segmentation on demineralized enamel.

### 2.2. Image Segmentation of OCT Dental Caries Frames

The data extraction algorithm and an associated designed graphic user interface (GUI) were written using the MATLAB (R2015a, MathWorks) programming language [47]. Our bespoke algorithm depends on a specific previously determined color map intensity threshold. This colormap can recognize early dental decay and distinguish three different types of dental caries tissue, which are *Intra*- (*Ia*) prismatic, *Inter*- (*Ie*) prismatic, and sound (healthy tissue) zones. The key idea for the identification of this colormap is based on finding a high correlation between OCT and SEM frames after similar frames are found between OCT and an automatic image segmentation for the SEM frames.

Figure 1 shows a schematic diagram of the scanning method and main steps of the segmentation algorithm. The image acquisition software Santec Inner Vision IVS-2000 and the OCT system were used to control the scan and choose the regions of interest (ROI) in the B-scan frames saved in “CSV” format. The detailed descriptions of segmentation steps are discussed in the following sections.

*a.* 
*Segmentation of enamel demineralization on SEM frames:*


The SEM frame segmentation can be summarized in the following steps, and Figure 2 shows the output frame for every processing to segment raw SEM frame of demineralized enamel: 1. The raw SEM frame (shown in Figure 2a) of a magnification of 500× with a pixel size of (1280 × 1100) is normalized to its maximum intensity, and the frame background is removed using multi-level image thresholds via Otsu’s method; the processed frame is shown in Figure 2b. 2. Enamel surface determination is performed by scanning the processed image by columns to find the first non-zero intensity pixel as a coordinate of the enamel surface. 3. A morphological filter with a disk-shaped structure with a 10-pixel diameter [48,49] is applied on the normalized frame (Figure 2b) and used to segment the *Intra*-prismatic and *Inter*-prismatic zones together, but not separately, by subtracting the processed image after the morphological filter (Figure 2c) from the normalized raw frame (Figure 2b), which results in a frame, as shown in Figure 2d, and the subtracted frame is binarized (Figure 2e). The filtered area corresponds to the *Intra*-prismatic and *Inter*-prismatic enamel with sample cracks and some noise. 4. *Intra*-prismatic zones can be separated from the normalized raw image (Figure 2b) by applying a scanning kernel window of (10 × 10) pixels in size with a function to count the pixel numbers, which have an intensity threshold of automatically selected quantized intensity level intervals along the enamel area. The automatically selected quantized intensity level intervals are determined using a quantized histogram for every one tenth of the maximum intensity, with the second highest peak of the quantized histogram corresponding to the automatically selected quantized intensity level intervals. The selected areas after applying the scanning kernel are the areas that have more than 25 pixels at each scanning kernel, as shown in (Figure 2g). 5. *Inter*-prismatic zones can be recognized by using MATLAB built in “Sobel” edge detection, which can demarcate only strong edges that are suitable to depict the *Inter*-prismatic enamel after applying image noise removal and finding the common areas between the two binarized frames of the processed Sobel edge (Figure 2f) and frame resulting after morphological filter (Figure 2e). The final segmented SEM frame is shown in Figure 2h with a red color label red for *Intra*-prismatic zone, yellow for *Inter*-prismatic zone, light blue for sound (healthy) tissue, and dark blue for the background. Figure 2**I** shows the enamel surface and two interfaces, one between the *Intra*-Prismatic and *Inter*-prismatic zones, and the other interface between the *Inter*-prismatic zone and sound tissue. The blue arrow indicates the normal thickness between the surface and *Intra*-prismatic/*Inter*-prismatic interface, and the red arrow indicates the normal thickness between the *Intra*-prismatic/*Inter*-prismatic interface and the *Inter*-prismatic/sound interface.

*b.* 
*Generating OCT color maps with receiver operating characteristic (ROC):*


In this process, the corresponding cropped area on the raw OCT frames and segmented SEM are superimposed. The cropped OCT frame has a much smaller pixel size than the SEM frame with the same ROI area. Therefore, the reconstruction of the cropped OCT frames is performed in the algorithm to compare the two frames of OCT and SEM with the same aspect ratio. ROCs curves are performed by collecting the intensity data of random pixels on OCT raw frames that correspond to the same area on the segmented superimposed SEM frames of the distinguished *Intra*-prismatic and *Inter*-prismatic zones; the areas under the ROC curves (AUC) for *Intra*-prismatic and *Inter*-prismatic zones are 0.9 and, 0.884, respectively, as shown in Figure 3.

The shortest distances of the nearest points on the ROC curve to the ideal point (sensitivity of 1 and (1-specificity) of 0) were measured to estimate the best color map thresholds for the two zones; the nearest points are (sensitivity of 0.797 and (1-specificity) of 0.17) for *Inter*-prismatic zone and (sensitivity of 0.791 and (1-specificity) of 0.136) for *Intra*-prismatic zone. These nearest points on the ROC curves correspond to color map thresholds of 29.7 and 20.7 dB for *Intra*-prismatic and *Inter*-prismatic zones, respectively, estimating the best color map thresholds for both zones to be used in measurements of the correlations between the OCT and SEM frames. For the best measurements of correlations, several color maps are used for trials of the best correlation. These color maps are slightly different but very close to the resulting color map thresholds from the best ROC estimation. The color map is only chosen based on the best correlation results between the OCT and SEM frames, and it can differ when using another OCT or SEM specifications for this correlation.

*c.* 
*Choosing the OCT colormap with the highest correlations to the corresponding segmented SEM frame:*


Frame correlations between OCT frame with a trial color map and a segmented SEM frame with the same corresponding mapping ROI are measured. The algorithm superimposed the two frames of the cropped OCT and SEM. If the two superimposed pixels are recognized with the same segmented zones, the logical result of the correlation is “true”; otherwise it is “false”, and the percentage of successive correlated area is then calculated for both zones and for the interfaces separating the *Intra*-prismatic and *Inter*-prismatic zones. Figure 4 shows an example of OCT/SEM correlation measurement for a test sample with three different ROIs (G1, G2, and G3). Cropped OCT frames with selected color map in column in Figure 4d is superimposed with SEM-corresponding frames of SEM in the column in Figure 4b, showing a good correlation at Figure 4g over all tested color maps, with the resulting correlation at Figure 4e,f. The correlation test was performed over the 8 prepared pre-demineralization samples, with a total of 24 regions. The results of the correlation show that for the *Inter*-prismatic zone, the maximum correlation percentage is 71%, the *Intra*-prismatic zone has a maximum correlation percentage of 86%, and their interfaces have a maximum correlation percentage of 99.9% with segmented SEM-corresponding frames. The selected OCT intensity color map will be used for our segmentation algorithm for demineralized dental image quantification. In our case, the selected intensity color map is designed with intensity ranges of 10–15.9 dB for the sound area, 16–27.9 dB for the *Inter*-prismatic (*Ie*) demineralization zone, and 28–60 dB for the *Intra*-prismatic (*Ia*) demineralization zone.

### 2.3. Labeling of the Inter- (Ie) and Intra-Prismatic (Ia) Demineralization Zones

For the determination of the enamel surface, the determination of the background (air in this case) was first conducted. The average reflectance value of the background (air) was obtained by testing and selecting random background regions with known areas of air above the surfaces of 35 B-scans extracted from the 8 specimens. It was found from the background intensity test that the maximum intensity recorded on the background can be used as a threshold of 5 dB to remove the background. Thereafter, the script locates and designates the first pixel in each depth-resolved reflectance line profile (A-scan) as an enamel surface.

Upon determining the surface, the algorithm then labels air, sound enamel, and the *Inter*- (*Ie*) and *Intra*- (*Ia*) prismatic demineralization zones with labeled colors based on the previously selected OCT intensity colormap defined in Section 2.2 and used for the segmentation of these different zones. In this case, the segmentation algorithm used the color red for the *Intra*-prismatic (*Ia*) demineralization zone, yellow for the *Inter*-prismatic (*Ie*) demineralization zone, blue for sound enamel, and dark blue for air (Figure 5a). The demineralization patterns of the 14–21 days’ caries lesions induced by the caries model used in this study presented distinct zones of *Inter*- (*Ie*) and *Intra*-prismatic demineralization (*Ia*), with the *Ia* zone limits located nearer to the surface and the *Ie* zone limits located deeper and closer to the *Ia* zone limits, as shown in Figure 5b. Figure 5a shows that the *Ia* zone is sandwiched by the *Ie* zone after segmentation; this is one case of the demineralization of the enamel degree. Another case of enamel demineralization is that the *Ia* zone is usually located near the surface, with few areas of the *Ie* zone near the surface. The *Ie* zones are located more after the *Ia* zones toward the enamel tissues and away from the surface, as shown in Figure 4d. In all cases, the Ia zone limits are located nearer to the surface, and the *Ie* zone limits are located deeper than the *Ia* zone limits.

### 2.4. Quantification of Mean Depths of the Inter- and Intra-Prismatic Demineralization Zones

Due to the slightly curved nature of the enamel surface, the mean depths of the *Inter*-($D_{Ie}$) and *Intra*-prismatic ($D_{Ia}$) demineralization zones were computed both with the original surface curvature and with the enamel surface aligned flat. There are two reasons to consider these two methods: The first reason is to investigate if the slight enamel surface curvature will affect the average depth measurements for each zone. The second reason is for the easy calculation of the average A-line starting from the enamel surface to the whole ROI. The surface-aligned frame will also be converted to a Microsoft Excel file format for manual calculation to verify the automated results.

In our case, the lateral and axial resolutions were 19.5 and 4.4 μm, respectively, and the cropped OCT frames had pixel sizes of 225 pixels laterally and 100 pixels axially. The 100 pixels in the axial direction were enough for our case to test a pre-caries demineralization to a maximum depth of more than 200 μm from the surface. The dimensional aspect ratio for the cropped OCT frames is calculated as 4.8 mm: 0.4 mm, which gives a sense of a slightly curvy enamel surface in our cropped frames. The curvature index of our samples, defined as a ratio between the enamel surface length and lateral frame length, ranged between 1.07 and 1.09. That means the two methods used to calculate the demineralization depth should be equivalent.

The OCT objective lens collects reflected and scattered beams from the nearly flat surface on the test sample. The collected beams will basically optically interfere with the reference OCT mirror and produce the OCT frame, which is a result of the Fast Fourier transform (FFT) of the combined optically interference signals of different subsurface levels.

In another case of OCT imaging, when the test sample has a strong curvature, approaches based on the tissue scattering intensities and beam direction make a difference between a flat surface and a curved surface, and an angular sample scan may be needed, as Memon et al. [50] used an angularly scanning sample arm with SS-OCT to scan a half-sphere sample for non-destructive metrology.

#### 2.4.1. Original Curved Surface

To measure the mean depths of the *Inter*- ($D_{Ie}$) and *Intra*-prismatic ($D_{Ia}$) demineralization zones of a slightly curved enamel surface, firstly, the tangent angles of each pixel of the identified enamel surface were determined using the “*skeletonOrientation*” MATLAB function (Figure 5a), and the median tangent angle α was calculated. The tangent angle is defined as the angle between the normal on the enamel surface and the axial direction. In our digital image analysis of the OCT frames for demineralized enamel, the enamel surface is defined by discrete pixels to represent the real continuous surface. The normal on the surface is estimated for each pixel on the enamel surface, which causes some error with a few pixels along the surface. To avoid these errors in determining the tangent angles on these few pixels, the algorithm considers a median instead of the average tangent angle, which enables the algorithm to select the most accurate tangent angle along the surface for the calculation of the mean demineralization depth. However, just using the average with the errors in tangent angles will result in an inaccurate tangent angle calculation because the median is not influenced by extremely large error values. With the median tangent angle, it was then possible to determine the location of the deepest pixel of the *Ie* and *Ia* zones along the normal of each pixel on the surface.

As shown in Figure 5b, a combined image of the original OCT file is overlaid with the determined enamel surface (in blue) and the limits of the *Intra*-prismatic (*Ia*) (in red) and *Inter*-prismatic (*Ie*) (in yellow) zones. These limits are determined as the last axial pixel of every segmented zone at every A-line (from the surface to the bottom of the frame on the enamel tissue).

The computation of the mean depths for the *Inter*- ($D_{Ie}$) and *Intra*-prismatic ($D_{Ia}$) demineralization zones are described in Equations (1) and (2) below.
(1)DIa=1n ∑i=1n(∆Ia)i×Psaxialcos⁡α
(2)DIe=1n ∑i=1n(∆Ie)i× Psaxialcos⁡αwhere i is the pixel order along the enamel surface, n is the total number of pixels along the enamel surface, α is the median of the tangent angle along the enamel surface, ${Ps}_{axial}$ is the axial pixel resolution, and ${\left(∆Ia\right)}_{i}$ and ${\left(∆Ie\right)}_{i}$ are the axial depths at pixel (i) for the Ia and Ie zones, respectively, measured from the enamel surface to the deepest pixel of each zone.

#### 2.4.2. With Aligned Surface

In this process, the average lesion depth can be calculated without any consideration of the enamel surface curvature. Figure 6 shows an example of the OCT selected raw frame (after background removal) and its enamel surface alignment to an axial level (dashed red line) corresponding to the highest axial level on the original enamel surface. To align the enamel surface, the algorithm shifts every A-line to the highest surface-aligned level, and then the color map intensity threshold is applied, and the average depth of every zone is calculated without surface angle consideration. Artificially flattening the curved surface would not counter the effect of angular scattering; however, the main purpose of using this aligned method for our slightly curved samples is to investigate a mean depth measurement depending only on the axial depth and to compare the results with the method of considering the slightly curved samples.

The computations of the mean depths for the *Inter*- (${DlI}_{e}$) and *Intra*-prismatic (${DlI}_{a}$) demineralization zones after surface alignment are described in Equations (3) and (4) below.
(3)$${DlI}_{a}=\frac{1}{n}\;\sum_{i=1}^{n}{\left(∆LIa\right)}_{i}\times {Ps}_{axial}\;\;$$
(4)$${DlI}_{e}=\frac{1}{n}\;\sum_{i=1}^{n}{\left(∆LIe\right)}_{i}\times {Ps}_{axial}\;\;$$where, i is the pixel number along the enamel surface, n is the total number of such pixels, ${Ps}_{axial}$ is the axial resolution, and ${\left(∆LIa\right)}_{i}$ and ${\left(∆LIe\right)}_{i}$ are the axial depths after enamel surface alignment at pixel i for the Ia and Ie zones, respectively, and measured from the enamel surface to the deepest pixel of each zone.

### 2.5. Algorithm Verification Using Simulated OCT B Scans

To verify the output from the algorithm described above and to determine its limitations in segmenting the demineralization zones and measuring the demineralization depth, the verification was conducted in two phases. In phase I, twenty-three synthetic OCT B-scans were generated to simulate demineralized enamel in various stages. The key idea of using simulated OCT B-scans of demineralized enamel was to uncover errors in the algorithm using simulated B-scans with predefined line profiles (A-scans) of demineralized enamel and with known mean depths of the *Inter*- ($D_{Ie}$) and *Intra*-prismatic ($D_{Ia}$) demineralization zones a priori.

The simulated B-scan required a predetermined A-scan and enamel surface profile. The simulated B-scan was generated by repeating the simulated A-scan laterally across the simulated enamel surface. These simulated OCT frames may not exactly approach the real OCT frame, as the imaging system is affected by many parameters of complicated enamel scattering and unexpected speckle noise; however, the OCT simulation is limited to our scope to check the suggested algorithm results, especially for the measurements of the mean demineralization depth.

To simulate the OCT line profile A-scans of demineralized enamel, three techniques were used:

Real average A-scan (RA_line), obtained from B-scans acquired from the specimens with known $D_{Ie}$ and $D_{Ia}$ values.Least-square-fit A-scan (LS_line), obtained by fitting an A-scan of an acquired B-scan with a polynomial degree of 20. The MATLAB function *polyval* was used for curve fitting.Mathematical A-scan (MA_line), obtained by using a combination of simulated mathematical functions, which can be simply approached to the real A-line signal.

A typical A-scan of demineralized enamel consists of a sharp rise from the background value in signal intensity at the enamel surface due to specular reflection, followed by some high backscatter intensities from the demineralized lesion zone, and then an exponential decay with an attenuation coefficient for *sound* enamel. The OCT A-scan has been described by others using the Beer–Lambert law [51], i.e.,
(5)$$I\left(z\right)\propto \;{exp}^{-2\mu_{t}z}$$where I is the reflected signal intensity, $\mu_{t}$ is the attenuation coefficient of the enamel, z is the axial depth measured from the enamel surface, and the number “2” refers to the round trip of light incidents on the enamel tissue and reflects to return and optically interfere with the OCT reference beam. The attenuation coefficient includes the absorption and scattering coefficient. Because the optical properties of teeth change after demineralization, the attenuation coefficients of *sound* and carious enamel differ [17]. The acquired A-scans of demineralized enamel are more complex than Equation (5). It can be described as the following mathematical approach with a trial to find another proportional coefficient, which can be used by changing its parameters to simulate a different expected A-line. The equation can be given as a combination of two peaks with an exponential decay as follows:(6)$$I\left(z\right)=I1\left(z\right)+I2\left(z\;[z1:z2]\right)$$
(7)$$I1\left(z\right)=az\;{exp}^{-2\mu_{t1}zS}+bz\;{exp}^{-2\mu_{t2}zS}\;\;\;\;\;\;\;\;\;$$
(8)$$I2\left(z\;[z1:z2]\right)=I1\left(z\;[z1:z2]\right)\times M\;\;\;\;\;\;\;\;\;\;\;\;\;\;\;$$where I1 describes the signal profile for combing demineralized and sound zones along all enamel until some depth z. a and b are the constants, S is a scale parameter (in this simulation, S=0.125), and $\mu_{t1}$ and $\mu_{t2}$ are the attenuation coefficients for *sound* and demineralized enamel, respectively. The second function I2 describes the signal profile in the demineralization zone between depths of z1 and z2 for the *Inter*-prismatic (*Ie*) demineralization zone. The scale parameter S is used to control the exponential damping of the simulated oct signal and then change the whole demineralization depth, while the magnification parameter M is used to control the reflection maximum intensity for the *Inter*-prismatic (*Ie*) demineralization zone. By changing the equation parameters, different A-line signals can be simulated with different demineralization depths of $D_{Ie}$ and $D_{Ia}$.

An extracted speckle noise should be added to the simulated B-scan using the mathematical A-scan (MA_line) and least-square-fit A-scan (LS_line); the frames simulated by the real average A-scan (RA_line) already have the speckle noise. To extract the speckle noise, the MATLAB median and Wiener filter can be used [52]. Figure 7 shows a median and Wiener filter applied on the original demineralized OCT frame with kernel windows of 3 × 3 and 5 × 5, respectively. The extracted speckle noise frames can be added to the simulated OCT frames.

To simulate the tooth surface, four different curvatures were used:A flat tooth surface (F_sur).The tooth surface from a real B-scan (S_sur).A curved tooth surface generated using a cosine function (M_sur).An irregular tooth surface drawn freehand by using MATLAB’s freehand function (D_sur).

Figure 8 shows examples of some simulated OCT frames of demineralized enamel used for algorithm verification before adding the extracted speckle noise.

In phase II of the verification process, twelve acquired B-scans from the 8 demineralized experimental prepared specimens were processed manually in a Microsoft Excel spreadsheet based on the same selected OCT intensity colormap and with the designed algorithm, and the extracted demineralization depth measurements were compared to determine the magnitude of deviations between the two measurements.

## 3. Results

For the code validation, 23 simulated OCT frames and 12 experimental OCT frames of demineralized enamel were processed using the suggested code. The findings of the comparison between the automated results and the manual calculations processed using a Microsoft Excel sheet resulted in strong correlations. The manual calculation processed using Excel for a tested frame was performed by the aligned frames converted to Excel format, and then the logarithmic values were converted to absolute intensity values, the average A-line intensity was calculated, and the average A-line intensity was converted again to logarithmic values. By using the same selected intensity color map threshold used for the automatic segmentation of demineralized zones, each zone can be determined, and the average depth of every zone can be calculated by counting the cells corresponding to every zone.

Figure 9 and Figure 10 show the correlation trendlines of the mean demineralization depth for simulated and real OCT frames, respectively. A strong correlation was found between the average demineralization depth resulting from the suggested code before the enamel surface alignment and the average demineralization depth resulting from the suggested code after the enamel surface alignment; the values of $R^{2}$ were calculated as 1 for each zone at the simulated and real OCT frames, as shown in Figure 9a,d and Figure 10a,d.

As for the Phase I verification process, comparisons of $D_{Ia}$ and $D_{Ie}$ of the 23 simulated B-scans were made between those computed manually and those computed by the algorithm both with and without surface alignment. The results of $D_{Ia}$ and $D_{Ie}$ from the aligned and original curved surfaces were almost identical (Figure 9a,d), and the $R^{2}$ value for the correlations between the manually computed and algorithm-computed $D_{Ia}$ and $D_{Ie}$ were 0.99 (Figure 9b,c) and 0.98 (Figure 9e,f), respectively.

As for the Phase II verification process, comparisons of $D_{Ia}$ and $D_{Ie}$ of the 12 acquired B-scans were made between those computed manually and those computed by the algorithm both with and without surface alignment. The results of $D_{Ia}$ and $D_{Ie}$ from the aligned and original curved surfaces were almost identical (Figure 10a,c), and the $R^{2}$ value for the correlations between the manually computed and algorithm-computed $D_{Ia}$ and $D_{Ie}$ were 0.97 (Figure 10b,c) and 0.99 (Figure 10e,f), respectively. The average difference between the value computed with the algorithm and the value computed manually for all 35 acquired B-scans is 2.6 um with a standard deviation of 3.3 µm. This result only means that there is high conformity between the automatic and manual calculations, but as an imaging principle of the spatial frequency range, with the sampling pitch, the absolute difference cannot be part of one pixel or have a value of less than an axial resolution of 4.4 µm.

## 4. Discussion

The segmentation of biological structures in OCT scans is a famous image processing technique used to either qualitatively classify the biological structures or quantitatively measure their geometry. An automated segmentation process would be especially invaluable to facilitate clinicians making objective diagnoses and to extract large amounts of quantitative volumetric data in longitudinal clinical trials.

The findings of this study showed that segmenting the two layers of demineralized enamel, the *Intra*-prismatic and *Inter*-prismatic zones, which were scanned with OCT for the early detection stage, may be possible. This finding with the suggested algorithm may be crucial for the speedy analysis of a huge amount of OCT frames, which helps to estimate the caries severity and provide a more accurate statistical assessment. In our study, a swept-source optical coherence tomography system (SANTEC IVS-2000 SS-OCT) with a central wavelength of 1300 nm was used for the 3D scanning of the demineralized enamel samples.

Our algorithm is a hybrid of global thresholding and multi-level quantized intensity threshold methods, where global thresholding was used to remove the background (air) and surface determination, and multi-level quantized intensity threshold was used to group the range of intensities to specific demineralization zones in the enamel [41]. An intensity threshold color map was built up by finding an image correlation between the scanning electron microscope and SS-OCT frames as a new strategy to use for OCT image segmentation and the quantification of dental enamel demineralization.

The segmentation by the determined intensity color map was used to measure the average depth of each demineralization zone. Two methods were used to calculate the average depth of each zone: considering the slight enamel surface curvature and using artificial surface alignment to check the effect of a slightly curved enamel on the average depth of each zone. The artificial caries lesions were created on the buccal surfaces of human-extracted premolars in this study, and the curvature of the buccal surfaces of these teeth is different than that found on incisors, lower premolars, and molars. Therefore, it was important to ascertain whether the use of the median tangent angle to compute $D_{Ia}$ and $D_{Ie}$ was appropriate and whether it was applicable for other curvatures. To determine whether the median tangent angle was appropriate, the mean depths of the *Inter*-$D_{Ie}$ and *Intra*-prismatic $D_{Ia}$ demineralization zones were computed both with the original surface curvature and with the enamel surface aligned flat. The results showed that the original curvature and the aligned surface were almost identical, indicating that using the median tangent angle was appropriate in the case of upper premolars. During the Phase I verification process, B-scans with different simulated surface curvatures were created from the A-scans of the pre-determined $D_{Ia}$ and $D_{Ie}$. The $D_{Ia}$ and $D_{Ie}$ outputs from the algorithm were like the pre-determined ones. This indicates its applicability to other cases of curvature.

Following the establishment of the auto-segmentation technique, but prior to its use for the extraction of large amounts of data, the systematic verification of the model or the full executable operations is critical to ensure that it does not demonstrate unintended behavior or harbor design errors. Verification is also necessary to assess its coverage, i.e., the extensiveness of its applicability. For this, verification was conducted in two phases. Two phases of the verification process were performed, i.e., against simulated and acquired real OCT scan frames of the B-scans of demineralized enamel, and we demonstrated that the results matched the expected output. In other words, the verifications demonstrated code–model equivalence.

The auto-segmentation method of demineralized enamel described here is applicable to demineralized enamel with distinct zones of *Inter*- and *Intra*-prismatic demineralization, therefore making a quantized intensity threshold segmentation of these zones possible. These distinct zones of demineralization are the result of the cyclic nature of demineralization subjected to the samples, the type and pH of the acid used, and the interval of demineralization. Different cyclic protocols, types, and pH levels of the acid and intervals of demineralization may produce different distributions of *Inter*- and *Intra*-prismatic demineralization. In conditions that produce demineralized enamel with less distinct *Inter*- and *Intra*-prismatic demineralization junctions, such as those in eroded enamel, the modification and revalidation of this segmentation technique may be necessary.

The image processing techniques used for the suggested algorithm of this study can be compared with a similar algorithm of a famous research work conducted by the old, famous, great school of dental image analysis used in OCT for the estimation of lesion severity by Professor Daniel Fried [16,18,27,28,30,32,33,36,53]. Regarding the edge-detection approach used in previously developed algorithms, the enamel edge and the lower lesion boundary were determined by applying an edge locator. The program first locates the maximum of each A-scan and differentiates the A-scan maximum as either demineralized or *sound* using the signal-to-noise ratio as a threshold. Another previous method used for edge detection is the anisotropic diffusion filter, which is designed to highlight boundaries, and finally, the threshold (Otsu) filter, which is used to identify the position of each edge by identifying the first pixels that fall under the threshold of e^−2^ of the maximum A-scan value. In the suggested algorithm of this study, an intensity threshold of 5 dB was applied to remove the background of the OCT raw frame, and enamel surfaces can be easily detected by scanning the frame for the first non-zero pixel. The locations of these pixels are the coordinates of the enamel surface. Then, the demineralization zone segmentation was applied to the raw OCT frame with the two discrete color zones of demineralization (red: *Intra*-prismatic; yellow: *Inter*-prismatic). The last pixel of each zone along each A-scan was considered the axial boundary of each demineralization zone. Therefore, this algorithm cannot only distinguish between sound enamel and demineralized enamel but also between each demineralization zone.

In previously developed algorithms, the depth of the lesion was calculated by locating the upper and lower edges using an edge-detection approach. The lesion severity was determined for each A-scan using two different approaches: fixed-depth and edge-detection approaches. In the fixed-depth approach, the enamel surface boundary was determined by the intersection of the line profile A-scan and a manually established straight line that marks the enamel surface in the A-scan. The lesion boundary was set to an arbitrary depth of 200 μm from the established enamel line. In the suggested algorithm of this study, the demineralization depth was also calculated using two approaches: without and with enamel surface alignment. In the approach without enamel surface alignment, the curvature nature of the enamel surface was considered in the calculation of the demineralization depth of every zone. In the approach with enamel surface alignment, the surface of the enamel was aligned to the highest point on the axes of the original enamel surface of the OCT frame, and every A-scan was shifted vertically to a new aligned surface level. In both techniques, the average demineralization depth for every demineralization zone was calculated, and all areas of the *Intra*-prismatic and *Inter*-prismatic zones can be calculated.

A bespoke algorithm was designed for the segmentation of the demineralization zone of the stack of cropped frames and was combined with a designed graphical user interface to help process a great amount of collected frames within a short time. The processing time estimated for one frame is less than 4 s, which is suitable for dental research demands. It is also demonstrated that the coverage of the model can be extended to various tooth curvatures. It is envisaged that this algorithm will greatly facilitate the study of the de- and remineralization kinetics of enamel caries lesions and eventually optimize remineralization therapy. Future applications of the algorithm could be used for more enhancements of deep learning segmentation algorithms for OCT scans in demineralization assessment and for the development of real-time oral OCT for clinical diagnostics of early demineralized enamel.

In conclusion, it was demonstrated that our multi-level quantized intensity threshold segmentation method for demineralized enamel described in this paper can be used to measure the *Inter*-and *Intra*-prismatic demineralization zones in artificial enamel caries, and we verified that there is code–model equivalence. This means that it can also calculate the occupied volume for every zone in a 3D OCT c-scan of demineralized enamel.

## Figures and Tables

**Figure 1 diagnostics-13-03586-f001:**
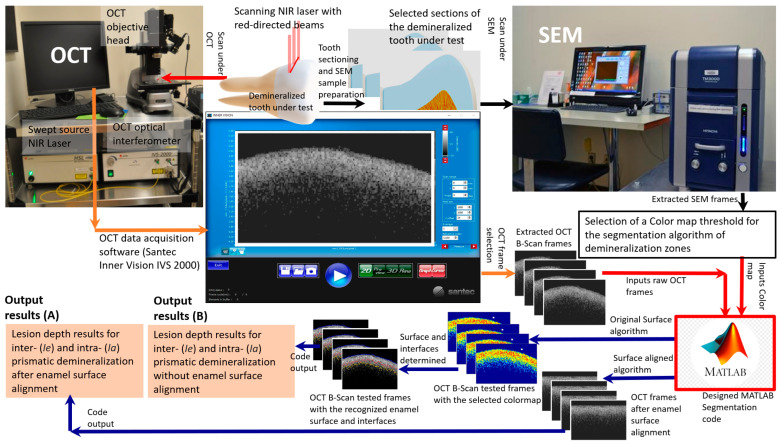
A schematic diagram of the scanning method and the main steps of the segmentation algorithm: the tooth with demineralization state was scanned under the OCT and B-scan frame collection at the ROI, and some selected sections were scanned under the SEM for SEM/OCT correlation and finding the colormap used for designing a segmentation code for the demineralized enamel. The code has two algorithms: original surface and surface alignment. The results are the lesion depth of *inter- Ie* and *intra- Ia* prismatic demineralization. SEM scans and correlation with OCT frames are needed for extracting the colormap for our OCT system. This colormap will be used for the segmentation of further OCT scans of the demineralized enamels, with no need to scan under the SEM.

**Figure 2 diagnostics-13-03586-f002:**
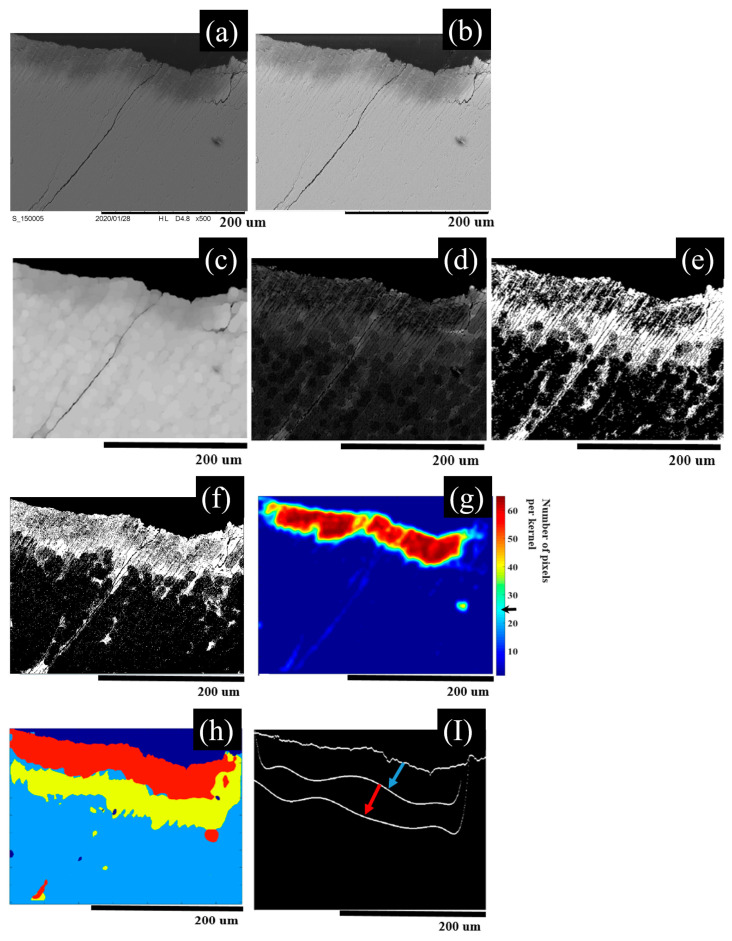
SEM segmentation of demineralized enamel: (**a**) raw SEM selected frame of demineralized enamel at 500×, (**b**) after intensity normalization and background removal, (**c**) after applying morphological disk filter, (**d**) after subtracting frame (**c**) from frame (**b**), (**e**) binary image conversion of frame (**d**). The filtered area corresponding to *Intra*-prismatic and *Inter*-prismatic enamel with sample cracks, (**f**) after Sobel edge detection, (**g**) after applying kernel of size (10 × 10) to count the number of pixels per kernel (color bar) with an auto-selected quantized intensity threshold. *Intra*-prismatic zone can be recognized by the area that has a number of pixels above 25 pixels per kernel indicated by the black arrow on color bar, (**h**) final segmented SEM frame with red color label red for *Intra*-prismatic zone, yellow for *Inte-r* prismatic zone, light blue for sound (healthy) tissue, and dark blue for the background. (**I**) Enamel surface and two interfaces, one between *Intra*-prismatic and *Inter*-prismatic zones, and other interface between *Inter*-prismatic zone and sound tissue. The blue arrow indicates the normal thickness between the surface and *Intra*-prismatic/*Inter*-prismatic interface, and the red arrow indicates the normal thickness between the *Intra*-prismatic/*Inter*-prismatic interface and *Inter*-prismatic/*sound* interface.

**Figure 3 diagnostics-13-03586-f003:**
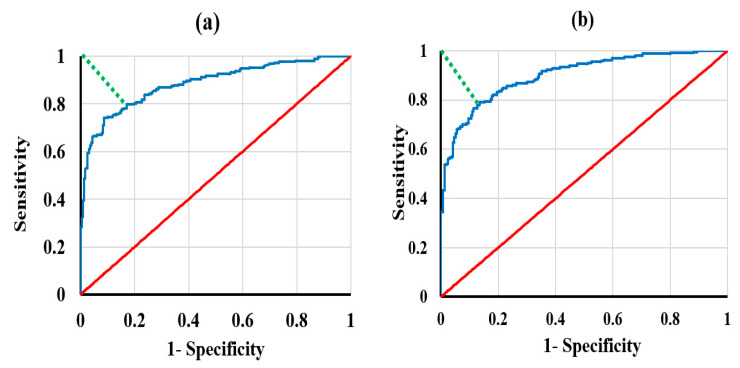
Receiver operating characteristic (ROC) curves (blue) of random pixel intensities on raw OCT frames corresponding to the same region of interest of the segmented SEM frames and baseline (red) (**a**) for *Intra*-prismatic zone and (**b**) for *Inter*-prismatic zone. Green dotted lines represent shortest distances of the nearest points on ROC curve to the ideal point (sensitivity of 1 and (1-specificity) of 0); the nearest points are sensitivity of 0.797 and (1-specificity) of 0.17 for *Inter*-prismatic zone and sensitivity of 0.791 and (1-specificity) of 0.136 for *Intra*-prismatic zone.

**Figure 4 diagnostics-13-03586-f004:**
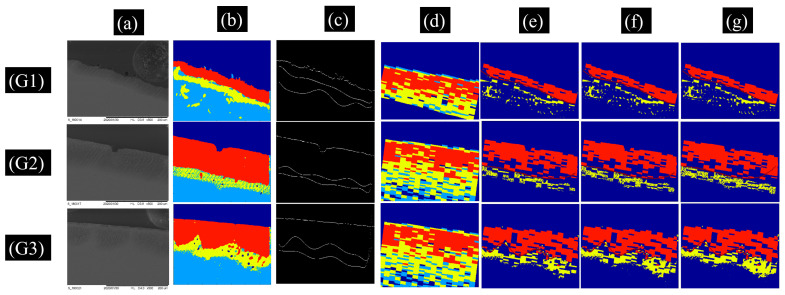
OCT/SEM frame correlation maps for a test sample with three ROIs (G1, G2, and G3). (**a**) Raw SEM images. (**b**) Segmented SEM (red:*Intra*-prismatic zone, yellow: *Inter*-prismatic zone, light blue: sound or healthy tissue, and dark blue: background). (**c**) Enamel surface and interfaces of *Intra*-prismatic and *Inter*-prismatic zones extracted from segmented SEM frame. (**d**) OCT-corresponding frames with a selected color map and segmented labels (red:*Intra*-prismatic zone, yellow: *Inter*-prismatic zone, light blue: sound or healthy tissue, and dark blue: background or the intensity below the colormap threshold). (**e**–**g**) Correlation results with three slightly different trail color maps (red: successive correlation at *Intra*-prismatic zone, yellow: successive correlation at *Inter*-prismatic zone, dark blue: sound or healthy tissue and background above the enamel surface). Column (**g**) is the best correlation by a selected color map on column (**d**).

**Figure 5 diagnostics-13-03586-f005:**
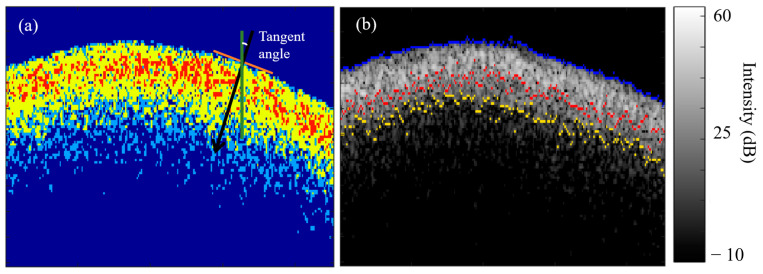
Quantification of mean depths of the *Inter*- ($D_{Ie}$) and *Intra*- prismatic ($D_{Ia}$) demineralization zones with the original surface curvature. (**a**) The *Intra*- prismatic (*Ia*) demineralization zone, *Inter*-prismatic (*Ie*) demineralization zone, sound enamel, and air were labeled red, yellow, blue, and dark blue, respectively. The red line is the tangent of one of the identified surface pixels. The tangent angle is the angle between the normal of the enamel surface (black arrow) and the vertical axis (green line). (**b**) A combined image of the original OCT file is overlaid with the determined enamel surface (in blue) and the limits of the *Ia* (in red) and *Ie* (in yellow) zones.

**Figure 6 diagnostics-13-03586-f006:**
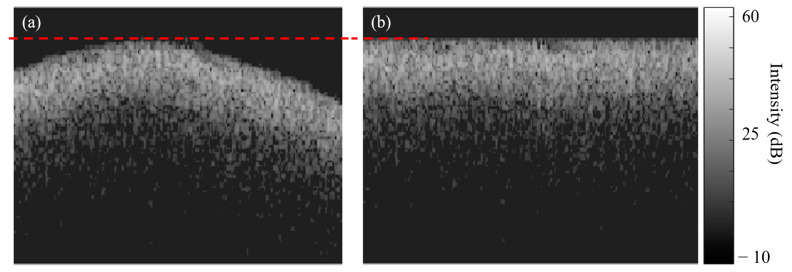
Enamel surface alignment. (**a**) Original B-scan showing surface curvature and (**b**) the same B-scan in (**a**) after enamel surface alignment to an axial level corresponding to the highest axial point (dashed red line) of the original enamel surface.

**Figure 7 diagnostics-13-03586-f007:**
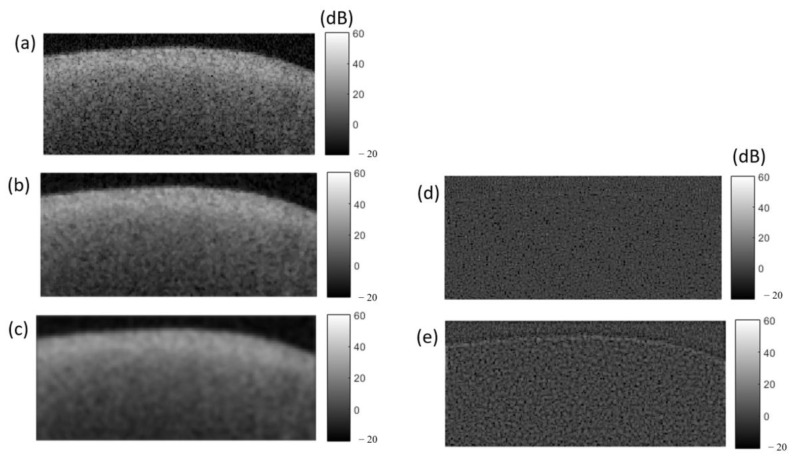
Subtraction of speckle noise in OCT. (**a**) Original B-scan of demineralized enamel via SANTEC (SS-OCT). (**b**) After applying MATLAB median filter. (**c**) After applying MATLAB Wiener filter. (**d**) Speckle noise extracted by subtracting median filter image (**b**) from the original image (**a**); (**e**) speckle noise extracted by subtracting Wiener filter image (**c**) from image (**a**).

**Figure 8 diagnostics-13-03586-f008:**
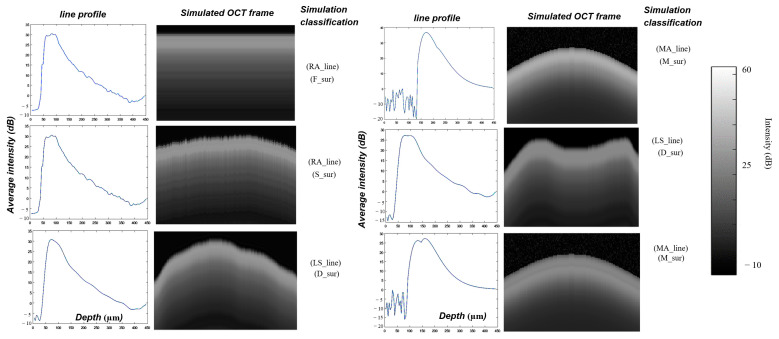
Examples of different simulated OCT line profiles and B-scans of demineralized enamel before adding extracted speckle noise for code validation. For A-scan simulation, (RA_line): real average A-scan; (LS_line): least-square-fit A-scan; and (MA_line): mathematical approach A-scan. For tooth surface curvature simulation, (F_sur): flat tooth surface; (S_sur): same surface as a selected real B-scan frame; (M_sur): mathematical equation surface; and (D_sur): freehand drawing surface.

**Figure 9 diagnostics-13-03586-f009:**
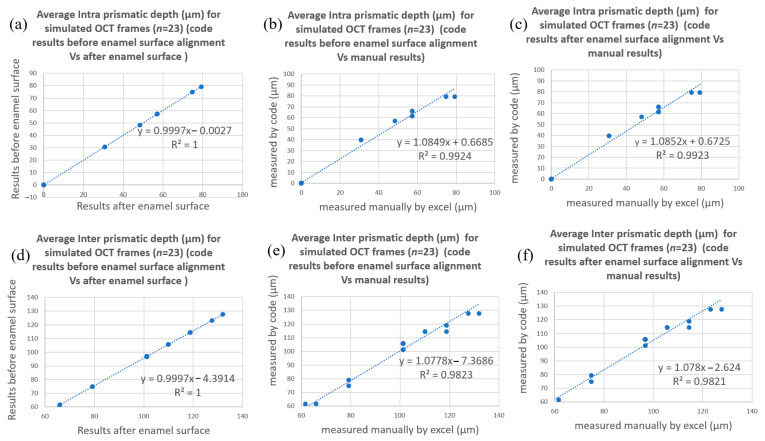
Correlation trendlines of the $D_{Ia}$ and $D_{Ie}$ of simulated OCT frames (*n* = 23). The results of $D_{Ia}$ and $D_{Ie}$ from aligned and original curved surfaces were almost identical (**a**,**d**), and the $R^{2}$ value for the correlations between manually computed and algorithm-computed $D_{Ia}$ and $D_{Ie}$ were 0.99, 0.98 (**b**,**c**) and 0.99, 0.98 (**e**,**f**), respectively.

**Figure 10 diagnostics-13-03586-f010:**
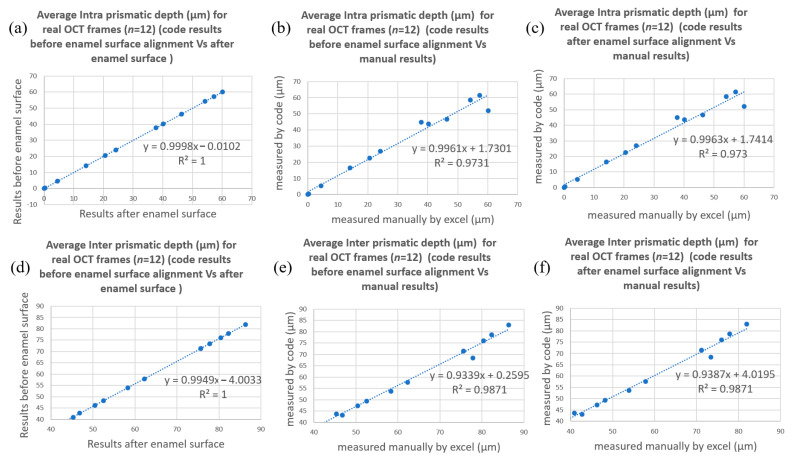
Correlation trendlines of the $D_{Ia}$ and $D_{Ie}$ of acquired OCT frames (*n* = 12). The results of $D_{Ia}$ and $D_{Ie}$ from aligned and original curved surfaces were almost identical (**a**,**d**), and the $R^{2}$ value for the correlations between manually computed and algorithm-computed $D_{Ia}$ and $D_{Ie}$ were 0.97 (**b**,**c**) and 0.99 (**e**,**f**), respectively.

## Data Availability

The data presented in this study are available in Appendix A.

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
