# Peer review of "Auto-Segmentation and Quantification of Non-Cavitated Enamel Caries Imaged with Swept-Source Optical Coherence Tomography"

_diagnostics, 2023, doi:10.3390/diagnostics13233586_

Round 1

Reviewer 1 Report

Comments and Suggestions for Authors

This work presents a method to measure the depth of an extracted tooth model's intra- and inter-prismatic demineralization zones (Die and Dia). The authors hope this method will greatly facilitate the study of de- and remineralization kinetics of enamel caries lesions and eventually optimize remineralization therapy. The authors first acquired OCT scans on some extracted human tooth models and then analyzed those images to quantify Die and Dia. To verify the quantification, the authors employed simulated B-scans in Phase 1 and actual sample images in Phase 2. They believe that the workflow they presented shows no error in Phase 1 and an error of 2.6 um with a standard deviation of 3.3 um in Phase 2.

Automated clinically relevant measurement would be critical for many medical applications, especially for those interoperative cases, where clinical decisions must be made within a limited time window. In the past decades, dental OCT has been actively developed and studied. A method to automate dental assessment by analyzing OCT images would be of great interest to both clinicians and imaging scientists.

Major concerns:

1.     The scientific contribution of this work is somewhat unclear. It appears that a previously published method was used for zone segmentation (ref 43). The innovation of this work seems to be merely the assessment of the depth of those segmented zones with Equations 1-4. If so, this limited contribution needs to be made clear to the audience.

2.     The overall description of the workflow is generally vague, and the technical details can not be fully assessed.

3.     The authors confused the resolution of an imaging system, which represents the spatial frequency range, with the sampling pitch (Lines 128 and 284). This is one of the most basic concepts in imaging science. This fundamental flaw needs to be corrected.

4.     The authors’ descriptions do not agree with the results. In Line 148, the authors state that the Ia zone is above the Ie zone. On the contrary, the images in Figure 2 show that the Ia zone was actually sandwiched by the Ie zone after segmentation. The entire section 2.3 does not explain this nor describe the segmentation detail. This makes the segmentation results not agree with the physiology the author stated.

5.     The Phase 1 results were reported as no errors. A detailed analysis with the actual data acquired, methodology used, and metrics compared is needed to justify the conclusion.

6.     If Equations 1-4 are the actual scientific contribution, they are overly simplified and need more rigor. The usage of the median tangent angle in Equations 1-2 is not justified. The formulation of Equations suddenly uses different symbols, DlIe and DlIa, without pointing out the meaning and the difference from Equations 1 and 2. Moreover, it needs to be clarified why the two methods were used or whether one is better. For the approaches based on the tissue scattering intensities, beam direction makes a difference – a flat surface fundamentally differs from a curved surface. Artificially flattening the curved surface would not counter the effect of angular scattering. A complete analysis with tissue light interaction and OCT principle is needed to compare the two methods.

7.     The Phase 1 of the authors’ verification has some significant problems. First of all, the description on MA_line is not clear. It is unclear whether Equations 6-8 are the content related to these A-line profiles. If so, modeling the line profile by a combination of functions is far from justified because tissue scattering and multiply scattering are generally complex to model. Thus, this phenomenological model could potentially misrepresent the actual images acquired. Furthermore, the speckle effect is completely ignored during the modeling. In particular, in Line 327, the authors mentioned that speckle reduction is not required. Yet, speckles are critical for intensity-based OCT image analysis, as they cause intensity fluctuations that can dramatically affect thresholding, segmentation, and clustering. 

8.     There needs to be more detail disclosed on the Phase 2 verification, and it was not justified. Therefore, this comparative verification with the manual method can not be well reviewed and assessed.

Other comments: 

1.     Figures 1, 4, and 5 are hard to read due to the limited resolution and small font size. A major improvement in their presentation is needed.

2.     Excessive keywords were used. Micro CT is irrelevant to the work.

3.     The sections of ‘introduction’ and ‘discussion’ are scattered, where a large portion of them discuss and summarize other’s works with little connection to the current paper established.

4.     The language in Lines 136-140 needs to be clarified, and the technical detail of the threshold selection can not be fully assessed.

5.     Line 291, the authors state that segmentation is the most important step for bioimage analysis. This is likely not true and subject to debate. If the authors believe so, the references are needed.

6.     English grammar error in Lines 154-157

Author Response

Author reply: Thank you so much for your valuable notes and helpful comments. The manuscript has been modified according to your valuable suggestions.

Please find my reply on all comments, The attached file is the modified manuscript

  1. The scientific contribution of this work is somewhat unclear. It appears that a previously published method was used for zone segmentation (ref 43). The innovation of this work seems to be merely the assessment of the depth of those segmented zones with Equations 1-4. If so, this limited contribution needs to be made clear to the audience.

Ans: Section (2.2. Image Segmentation of OCT Dental Caries Frames) has been added in details about our segmintation algorithm and how we obtained the color map from OCT/SEM correlation. With more clarification of the missing points. The innovation of this work is the design of an image processing algorithm for demineralization zones and the average demineralization depth information along the demineralized enamel. (ref 43) was briefly described as the method; however, this manuscript gives more details about the designed Algorithm with a verification method.

  1. The overall description of the workflow is generally vague, and the technical details cannot be fully assessed.

Ans: Thanks for your comment. This manuscript now has a clearer workflow as it presents a new idea for finding a colormap threshold by OCT/SEM correlation, then applies this colormap for fast and simple multi-level intensity threshold segmentation of two types of enamel demineralization, then verifies the algorithm outputs by simulated and real OCT frames and discusses the importance of the work and the applications. If possible, let us know if other unclear points need more description.

  1. The authors confused the resolution of an imaging system, which represents the spatial frequency range, with the sampling pitch (Lines 128 and 284). This is one of the most basic concepts in imaging science. This fundamental flaw needs to be corrected.

Ans: The mistake was corrected on (lines 581-586) “The average difference between the computed with the algorithm and manually for all 35 acquired B-scans is 2.6 um with a standard deviation of 3.3 µm. This result only means the high conformity between the automatic and manual calculation, but as an imaging principle of the spatial frequency range, with the sampling pitch, the absolute difference can’t be part of one pixel or axial resolution of 4.4 µm”.

  1. The authors’ descriptions do not agree with the results. In Line 148, the authors state that the Ia zone is above the Ie zone. On the contrary, the images in Figure 2 show that the Ia zone was actually sandwiched by the Ie zone after segmentation. The entire section 2.3 does not explain this nor describe the segmentation detail. This makes the segmentation results not agree with the physiology the author stated.

Ans: The correction has modified at lines (311-317), the figure number has changed from figure 2 to figure 5. Section 2.2 now explains the segmentation algorithm in detail.

“Figure 5a shows that the Ia zone is sandwiched by the Ie zone after segmentation; this is one case of the demineralization of the enamel degree. Another case of enamel demineralization is that the Ia zone is usually located near the surface, with few areas of the Ie zone near the surface. The Ie zones are located more after the Ia zones toward the enamel tissues and away from the surface, as shown in Fig. 4d. In all cases, the Ia zone limits are located nearer to the surface, and the Ie zone limits are located deeper than the Ia zone limits.”

  1. The Phase 1 results were reported as no errors. A detailed analysis with the actual data acquired, methodology used, and metrics compared is needed to justify the conclusion.

Ans: A detailed analysis of the actual data has been added in Section 3 and Figure 9.

  1. If Equations 1-4 are the actual scientific contribution, they are overly simplified and need more rigor. The use of the median tangent angle in Equations 1–2 is not justified. The formulation of Equations suddenly uses different symbols, DlIe and DlIa, without pointing out the meaning and the difference from Equations 1 and 2. Moreover, it needs to be clarified why the two methods were used or whether one is better. For the approaches based on the tissue scattering intensities, beam direction makes a difference – a flat surface fundamentally differs from a curved surface. Artificially flattening the curved surface would not counter the effect of angular scattering. A complete analysis with tissue light interaction and OCT principle is needed to compare the two methods.

Ans: Equations 1-4 are now described in more detail; the reason to using the two methods are now clarified section 2.4. Due to a slight enamel surface curvature and dimensional aspect ratio, median tangent angle can be suitable for mean depth calculations.

“There are two reasons for considering these two methods: first, to investigate if the slight enamel surface curvature will affect the average depth measurements for each zone. The second reason is for easy calculation of the average A-line starting from the enamel surface to the whole ROI. The surface-aligned frame will also be converted to Microsoft Excel file format for manual calculation to verify the automated results.

Equations 1-4 are just two methods used to calculate the average depth of each zone at the final stage, not the main scientific contribution. However, the key contribution is finding a colormap intensity threshold that can be used for image segmentation of the two different demineralization zones. We added more details about this key idea of this algorithm.”

  1. Phase 1 of the authors’ verification has some significant problems. First of all, the description on MA_line is not clear. It is unclear whether Equations 6-8 are the content related to these A-line profiles. If so, modeling the line profile by a combination of functions is far from justified because tissue scattering and multiply scattering are generally complex to model. Thus, this phenomenological model could potentially misrepresent the actual images acquired. Furthermore, the speckle effect is completely ignored during the modeling. In particular, in Line 327, the authors mentioned that speckle reduction is not required. Yet, speckles are critical for intensity-based OCT image analysis, as they cause intensity fluctuations that can dramatically affect thresholding, segmentation, and clustering. 

Ans: More details, clarifications and symbol definitions have been added. Equations 6-8 are introduced under the description of (MA-line) or the A-line simulated only by the mathematical approach. The simulation by mathematical equation was just an approach to creating a line profile similar to the real one, the key is to verify that the outputs of the known average depth simulated OCT B-scan near the real frame but do not simulate the exact OCT B-scan which contains more complex scattering and speckle noise. However, there is another method to simulate the A-line with a real line profile. The speckle pattern frame has been extracted and added to the mathematically simulated frame; the results didn’t affect the average demineralization depth for each zone.

At the end of the introduction, speckle is more widely discussed and the reason of not using a speckle removal has been discussed. Thanks for the valuable feedback.

  1. There needs to be more detail disclosed on the Phase 2 verification, and it was not justified. Therefore, this comparative verification with the manual method can not be well reviewed and assessed.

Ans: Figure 9 and 10 with Section 3 and more details about manual calculation have been added.

Other comments: 

  1. Figures 1, 4, and 5 are hard to read due to the limited resolution and small font size. A major improvement in their presentation is needed.

Ans: Done, thanks. (All figures are saved in a PPT file, but I can’t upload unless there is one file.)

  1. Excessive keywords were used. microCT is irrelevant to the work.

Ans: Done.

  1. The sections of ‘introduction’ and ‘discussion’ are scattered, where a large portion of them discuss and summarize other’s works with little connection to the current paper established.

Ans: Modifications are made in the introduction and discussion. Now the manuscript is focused more on our algorithm and OCT.

  1. The language in Lines 136-140 needs to be clarified, and the technical detail of the threshold selection can not be fully assessed.

Ans: Modified.

  1. Line 291, the authors state that segmentation is the most important step for bioimage analysis. This is likely not true and subject to debate. If the authors believe so, references are needed.

Ans: Modified.

  1. English grammar error in Lines 154-157

Ans:

All Sections have been rewritten.

Please let me know if you have further questions.

Thank you for your valuable review of our manuscript.

Reviewer 2 Report

Comments and Suggestions for Authors

In this article, the authors present an algorithm that contributes to the application of oct imaging technique in the segmentation and measurement of Non-cavitated Enamel Caries Imaged. The depth of demineralization in inter- (DIe) and intra-prismatic (DIa) demineralized areas was measured by data from the pair-written algorithm. The model and idea of this algorithmic measurement is very novel and effective. However, I think the article still needs to be improved before publication. Here are my comments and questions:

1. the main purpose of the article is to introduce the depth measurement algorithm. However, the article is more about comparing and analyzing the data obtained from the algorithm, so I suggest that the authors can introduce and discuss more about the content of the algorithm.

2. The 8 teeth measured in the experiment were maxillary premolars without obvious caries plus the demineralization depth data of inter- (DIe) and intra-prismatic (DIa) demineralized areas artificially demineralized in different environments. I suggest that the authors can add 2 teeth after natural demineralization for comparative measurements, which will increase the authority of the experiment.

3. The author's article only proves the feasibility of the algorithm, I think the author can also introduce the advantages and convenience of the algorithm. This will better illustrate the role of this algorithm in the study of demineralization and remineralization kinetics of enamel caries lesions and in achieving optimization of remineralization treatment. 

Author Response

Author reply: Thank you so much for your valuable notes and helpful comments. The manuscript has been modified according to your valuable suggestions.

  1. The main purpose of the article is to introduce the depth measurement algorithm. However, the article is more about comparing and analyzing the data obtained from the algorithm, so I suggest that the authors introduce and discuss more about the content of the algorithm.

Ans: Thank you for your valuable suggestion. The flow of this work is to describe a designed algorithm based on an easy idea for image segmentation of demineralized enamel zones by OCT/SEM frame correlation and finding a thresholding intensity colormap. Then the method of using segmentation to measure the demineralization depth has been described. At the final stage, algorithm verification by simulated and real OCT has been performed. Also, the advantages and future applications have been discussed.

Section (2.2. Image Segmentation of OCT Dental Caries Frames) has been added in details about our segmintation algorithm and how we obtained the color map from OCT/SEM correlation. With more clarification of the missing points. The innovation of this work is the design of an image processing algorithm for demineralization zones and the average demineralization depth information along the demineralized enamel. (ref 43) was briefly described as the method; however, this manuscript gives more details about the designed algorithm with a verification method.

Also, the introduction and discussion of this manuscript have been modified to focus more on the suggested algorithm. If possible, let us know if other unclear points need more description.

  1. The 8 teeth measured in the experiment were maxillary premolars without obvious caries plus the demineralization depth data of inter- (DIe) and intra-prismatic (DIa) demineralized areas artificially demineralized in different environments. I suggest that the authors add 2 teeth after natural demineralization for comparative measurements, which will increase the authority of the experiment.

Ans: Ans: This is a good suggestion. Thank you so much. It will add more value to the work. However, it is now impossible to add other teeth after natural demineralization or do other experimental scans because the project has ended, and some authors are now away from the lab or going back to their home country. But it can be considered in our future work.

  1. The author's article only proves the feasibility of the algorithm; I think the author can also introduce the advantages and convenience of the algorithm. This will better illustrate the role of this algorithm in the study of demineralization and remineralization kinetics of enamel caries lesions and in achieving optimization of remineralization treatment. 

Ans: Thanks again for your valuable suggestions. The advantages of the algorithm have been stated, especially at the end of the discussion.

Please let me know if you have further questions.

Thank you for your valuable review of our manuscript.